# Geospatial Analysis of Environmental Atmospheric Risk Factors in Neurodegenerative Diseases: A Systematic Review

**DOI:** 10.3390/ijerph17228414

**Published:** 2020-11-13

**Authors:** Mariana Oliveira, André Padrão, André Ramalho, Mariana Lobo, Ana Cláudia Teodoro, Hernâni Gonçalves, Alberto Freitas

**Affiliations:** 1CINTESIS—Center for Health Technology and Services Research, Faculty of Medicine, University of Porto, Rua Doutor Plácido da Costa, s/n, 4200-450 Porto, Portugal; andrelcramalho@gmail.com (A.R.); nanalobo@gmail.com (M.L.); hernanigoncalves@med.up.pt (H.G.); alberto@med.up.pt (A.F.); 2Department of Community Medicine, Information and Health Decision Sciences (MEDCIDS), Faculty of Medicine, University of Porto, Rua Doutor Plácido da Costa, s/n, 4200-450 Porto, Portugal; 3Faculty of Arts and Humanities, University of Porto, Via Panorâmica, s/n, 4150-564 Porto, Portugal; andre27.f@hotmail.com; 4Department of Geosciences, Environment and Land Planning, Faculty of Sciences, University of Porto, Rua do Campo Alegre 687, 4169-007 Porto, Portugal; amteodor@fc.up.pt; 5Earth Sciences Institute (ICT), Pole of the FCUP, University of Porto, 4169-007 Porto, Portugal

**Keywords:** neurodegenerative, environment, geospatial, epidemiology, systematic review

## Abstract

Despite the vast evidence on the environmental influence in neurodegenerative diseases, those considering a geospatial approach are scarce. We conducted a systematic review to identify studies concerning environmental atmospheric risk factors for neurodegenerative diseases that have used geospatial analysis/tools. PubMed, Web of Science, and Scopus were searched for all scientific studies that included a neurodegenerative disease, an environmental atmospheric factor, and a geographical analysis. Of the 34 included papers, approximately 60% were related to multiple sclerosis (MS), hence being the most studied neurodegenerative disease in the context of this study. Sun exposure (*n* = 13) followed by the most common exhaustion gases (*n* = 10 for nitrogen dioxide (NO_2_) and *n* = 5 for carbon monoxide (CO)) were the most studied atmospheric factors. Only one study used a geospatial interpolation model, although 13 studies used remote sensing data to compute atmospheric factors. In 20% of papers, we found an inverse correlation between sun exposure and multiple sclerosis. No consensus was reached in the analysis of nitrogen dioxide and Parkinson’s disease, but it was related to dementia and amyotrophic lateral sclerosis. This systematic review (number CRD42020196188 in PROSPERO’s database) provides an insight into the available evidence regarding the geospatial influence of environmental factors on neurodegenerative diseases.

## 1. Introduction

The human brain suffers pathological changes in the process of ageing, leading to a range of neurodegenerative disorders [1]. Until 2050, the world’s population over 60 years old is expected to nearly double compared to 2015, reaching 22% [2]. The ageing of the population raises concerns about the future of neurodegenerative diseases [3] and urges the development of better health systems with long-term care tailored to meet the needs of the elderly population [2]. These diseases are characterised by progressive damage or loss of neuronal cells, leading to compromised functioning of the brain, with either motor or cognitive roles being endangered [4]. Alzheimer’s disease (AD) is the most prevalent neurodegenerative disease worldwide [5], and the most common cause of dementia and it is a progressive disease that can be characterised by extensive cognitive damage, commonly affecting the functional competency to perform quotidian actions [6]. Following AD in prevalence, Parkinson’s disease (PD) is defined by the dopaminergic neuronal loss, which presents as a Lewy Bodies (LB) disease that during the course of the disease spreads to different regions of the brain, leading to motor and non-motor symptoms [7]. The third most prevalent neurodegenerative disease is multiple sclerosis (MS), affecting around 2.5 million people worldwide, which is an inflammatory demyelination disorder of the brain, frequently of autoimmune origin [5].

Apart from ageing, the most common risk factors in the development of neurodegenerative diseases include gender, clinical history of hypertension, diabetes, cranial injury, tumours, and infections, as well as smoking and drinking habits [8]. However, the knowledge on the development of most of these diseases remains incomplete, including the influence that environmental factors might have in this context [8]. This review focuses on atmospheric pollution due to its significant impact worldwide, with 91% of the world’s population living under air pollution levels exceeding the established limits by the World Health Organisation (WHO) [9]. Atmospheric pollutants are all “substances, laid by the human activities with enough concentration to cause detrimental influences to health (…)” [10], but some authors also consider natural sources of pollutants [11].

Recent studies point out that atmospheric pollution can trigger mechanisms responsible for neurodegenerative diseases, especially particulate matter with a diameter under 2.5 µm (PM_2.5_) due to its capacity to reach the brain [12]. Four main pathways have been identified to air pollution reaching the brain [13]: systemic inflammation (in which the pollutant penetrates peripheral organs such as the lungs, provoking a systemic response that transfers the inflammation to the brain) [13,14,15,16,17], nasal olfactory (inhaled ultrafine and fine particles that penetrate the lungs reaching the systemic circulation, the trigeminal nerve, the brainstem, the hippocampus, and the frontal lobe) [18,19,20,21,22,23], adsorbed compounds (particulate matter from air pollution encloses various toxic compounds that may be adsorbed independently of the particle) [24,25,26] and inhaled oxidation (reactive oxygen agents such as ozone, which interact with proteins and lipids in the lungs altering them into toxic compounds that provoke oxidative stress, leading to various brain damages) [27,28,29,30,31]. PM_2.5_ is associated with AD, PD, dementia, autism, and stroke [32]. Other pollutants such as ozone (O_3_), particulate matter with a diameter under 10 µm (PM_10_), nitrogen dioxide (NO_2_), carbon monoxide (CO), and sulphur dioxide (SO_2_) are also toxic to the central nervous system (CNS) [33]. Despite the abundance of studies on neurodegenerative diseases and the environmental influence on the CNS, studies considering a geospatial approach are scarce, and many of them only consider the geographic area as a clustering factor [34,35,36], not proceeding to further analysis.

One of the tools for environmental data extraction and analysis is remote sensing. It has been broadly used in epidemiological studies, increasing the number of publications featuring remote sensing applied to health from 5.6% in 2007 to 13.3% in 2016 [37]. Apart from remote sensing, GIS (geographic information systems), clustering, and spatial models are other tools that can be used to analyse environmental data. Geospatial analysis is a broad concept that combines both exposure assessment methods and posterior statistical analysis, allowing us to integrate information on health and disease of specific regions with environmental data as well as other spatially distributed relevant data such as socio-economic information. Understanding the strengths and limitations of previous studies that used geospatial approach could inform us on how to better delineate future research, thereby maximising the gains regarding the understanding of the geospatial distribution of environmental risk factors associated with neurodegenerative diseases. This type of knowledge is important to establish to aid in planning future policies that mitigate environmental factors, which may be critical in the prevention and progression of neurodegenerative diseases.

It is important to frame the issue of neurodegenerative diseases in the new coronavirus pandemic caused by SARS-CoV-2 (Severe Acute Respiratory Syndrome Corona Virus-2). This virus can penetrate the CNS mainly through two pathways: by retrograde axonal transport via the cribriform plate, and by the systemic circulatory system [38]. This last pathway is similar to one of the atmospheric pollutants reaching the CNS, raising the hypothesis that this virus may pose a higher potential for neuroinflammation and neurodegeneration [39]. The concept of Neuro-COVID-19 is, thus, being increasingly used in current research [40].

In this systematic review, we aim to identify studies concerning neurodegenerative diseases and their environmental atmospheric risk factors, which have used geographical analysis/tools as part of their work. The methods used to conduct this systematic review are presented in Section 2 (Materials and Methods), including the query used, the inclusion and exclusion criteria, and the study characteristics gathered by the reviewers. In Section 3 (Results), there are two subsections, the first one being identification, screening, and assessment (which explains the main numbers retrieved by the systematic review, from the total number of studies screened to the included papers and their characteristics) and the second being qualitative synthesis (in which the main relations retrieved between the environmental and neurodegenerative diseases are shown). The results from the systematic review are discussed and compared to those in the literature in Section 4 (Discussion), and study limitations are explored. To finalise, a small conclusion is presented in Section 5.

## 2. Materials and Methods

We searched PubMed, Web of Science, and Scopus databases from inception to the 4th February 2020, date in which the last search was run. The oldest study retrieved was from 1951 [41]. All studies with at least one neurodegenerative disease, one atmospheric pollutant or factor, and some geographic approach were included for screening, with a total of 4655 abstracts screened. No papers were excluded due to language restrictions. For the literature review, we screened the databases using the following keywords: [((alzheimer*) OR (ataxia*) OR (Chorea Minor) OR (creutzfeldt*) OR (dementia*) OR (Frontotemporal) OR (Guillian-barre syndrome) OR (Huntington*) OR (kennedy* disease) OR (Lewy*) OR (motor neuron*) OR (Myotonic dystrophy) OR (neurodegen*) OR (parkinson*) OR (pick’s) OR (Prion) OR (progressive AND palsy) OR (progressive muscular atrophy) OR (sclerosis*) OR (*senile) OR (spinal atrophy)) AND ((atmospher*) OR (carbon) OR (environment*) OR (humidity) OR (meteorologic*) OR (nitrogen) OR (ozone) OR (particulate*) OR (PM2*) OR (pollut*) OR (sulphur*) OR (surface pressure) OR (temperature)) AND ((drone) OR (geograph*) OR (imagery) OR (landsat) OR (map) OR (mapping) OR (modis) OR (remote sens*) OR (satellite) OR (sentinel) OR (spatial) OR (topologic*))]. This query was a result of the sensitivity analysis of terms that could be relevant for this review, keywords that added no papers to the results were excluded from the query. These included the terms: geospatial, spatio*, and cartograph*. This review was submitted to PROSPERO under the registration number CRD42020196188.

All studies with at least one neurodegenerative disease, one atmospheric pollutant or factor, and some geographic approach were included for screening. The inclusion criteria were the following: (1) Studying a neurodegenerative disease; (2) Accounting for atmospheric environmental factors or pollutants; (3) Using geospatial analysis or tools; (4) All the previous criteria in the same study. Studies were excluded if: no evidence of geographic analysis was found (stating a geographic area without further analysing it or comparing it to other areas was an exclusion motive), no atmospheric pollutants or factors referred (soil and water pollutants were excluded from this revision) and no focus on neurodegenerative diseases (studies about the mechanisms and biospecimen behind the disease were discarded). Full papers were independently read by the same researchers to apply the inclusion and exclusion criteria further. Unavailable full texts were requested to the authors individually. Those studies that remained unavailable were excluded.

For the included full papers data extraction and analysis, structured forms were created and used by the reviewers. The forms included free writing inputs such as the title, year, country, authors, DOI (digital object identifier), participants, aims, and key findings. Additionally, multiple-choice inputs were available for study type, statistical methods, outcome measurements, study limitations, neurodegenerative disease, environmental risk factors, geographic approach, and type of geographical approach. A level of simplification was applied to categorise the study limitations, to better fit most studies on the main biases and issues encountered. Both reviewers filled these forms independently and discussed the extracted information at the end of the process. Nonetheless, the reviewers had the option to input free text if they considered necessary to complement any of the inputs mentioned. Risk of bias from individual studies was assessed by fitting the study limitations into one or more of the pre-defined form options: none given by the authors; conflict of interests (any conflict of interests stated by the authors); confounding factors (unassessed confounding factors); ecological bias (extrapolation of a conclusion from a population to a patient); exposure assessment (issues with assessing the patient’s exposure to the environmental factor); interpolation (issues with spatially interpolating the environmental factor); recall bias (data collection relying on patient’s memory); migration of patients (patients moving from one residence to another); referral bias (studies relying on the doctor’s referring similar cases); sampling issues (under sampling; over sampling or non-representative sampling); statistical issues (lacking of more relevant statistical methods); study design issues (studies acknowledging inappropriate study design); survival bias (relying on a patient being alive over a period of time); time related issues (inability to assess the amount of time a patient was exposed to the environmental factor); unassessed patients (patients outside the databases not being considered).

## 3. Results

### 3.1. Identification, Screening, and Assessment

Of the 4655 articles initially screened, only 34 were included in the final study. The selection process is summarised using PRISMA (Preferred Reporting Items for Systematic Reviews and Meta-Analyses) [42] presented by a flow diagram (Figure 1). In the diagram of Figure 1, it is possible to identify that 4655 records were screened in this systematic review, most of which did not relate simultaneously to neurodegenerative diseases, atmospheric factors, and geographical factors (*n* = 3239). After screening, 90 full papers were assessed for eligibility, from which nearly half (*n* = 44) did not tackle any atmospheric pollutants or factors. By the end of the selection process, 34 articles were included. The discrepancy in the total number of excluded studies and the sum of the exclusion reasons categories is due to the overlapping of exclusion reasons.

The number of times each disease, environmental factor, and geographical approach was retrieved are detailed in Table 1. The sum of the categories’ absolute frequencies for environmental and geographic factors exceeds the number of studies because they are not mutually exclusive. Most papers focused on (MS) (56%), of which 63% studied sun exposure as the main environmental factor, mostly using remote sensing data (58%), always combined with another geographic factor. Administrative divisions were the most used geographical approach when considering all diseases (47%).

The main characteristics of the studies—study design, outcomes, statistical methods, and study limitations—are summarised in Table 2. Most of the studies are cross-sectional (*n* = 12) and correlation was the most reported effect measure (*n* = 19) and statistical method (*n* = 22) used. The most common study limitations found were related to exposure assessment (*n* = 18) and confounding factors *n* = 15). The full characteristics of the studies can be found in Table 3. This table contains all the characteristics retrieved by the reviewers for each included study: country of origin, year of publication, neurodegenerative disease, environmental factor, geographic factor, study design, study limitations, statistical methods, and outcome. A list of excluded studies [1,7,43,44,45,46,47,48,49,50,51,52,53,54,55,56,57,58,59,60,61,62,63,64,65,66,67,68,69,70,71,72,73,74,75,76,77,78,79,80,81,82,83,84,85,86,87,88,89,90,91,92,93,94,95,96] is also available in Table A1, Appendix A.

The country with the highest number of published papers under the criteria of this systematic review was the United States of America (USA), with nine included papers. It was followed by Canada with five papers. However, Europe, with the aggregation of articles from Bulgaria, England, France, Italy, Norway, and Spain totals twelve included papers. The Middle East and Asia had three papers, and Australia had two papers. Table 4 contains the papers studied by each geographical region, grouped by neurodegenerative disease.

Of the 34 included papers, 19 were published in the last 5 years. Until 2000, only four studies were found corresponding to the criteria of this systematic review, all of which focused on multiple sclerosis. Until 2007, except for a paper focusing on Parkinson’s disease, no other neurodegenerative disease had been considered.

### 3.2. Qualitative Synthesis

Among the 19 papers focused on MS, with sun exposure being the most studied environmental factor (*n* = 13), one was methodological and did not measure any outcome related to MS [97]. An inverse association between MS and sun exposure was reported in seven studies [98,99,100,101,102,103,104]. In contrast, in the other three studies, no significant association was found [105,106,107], one of which attributed the results to the adjusting for latitude [105].

As for temperature, five studies found it to be negatively related to MS [98,99,100,103,108] and one found this association to be due to its high correlation with latitude [105]. Precipitation was found to be uncorrelated [98,100,105] or only weakly positively correlated [99,108] to MS. Humidity was focused only on two studies, both of which found no correlation to MS [98,105].

Concerning the environmental pollutants, PM_2.5_ was found to be positively correlated with MS in one study [109] and not significantly correlated with another [110]. SO_2_ had no correlation [110,111] except in one study [112]. PM_10_ and NO_X_ (nitrogen oxides) were studied only twice and studies disagree in the reported correlation [110,112]. Additionally, a study found a positive association between an air quality index (AQI) and MS [113], and magnesium (Mg) was found to be negatively correlated [114], while radon was not found to be associated with MS [115]. In the domain of paediatric MS, PM_10_ was found to be related to it [116,117], as well as PM_2.5_, Pb (lead), SO_2_, and CO [117].

Of the seven studies concerning Parkinson’s Disease (PD), Pb was found to be positively spatially related [118,119], as well as Cu (copper), Mg [118], and Mn (manganese) [120]. PM_10_ and PM_2.5_ were not correlated [121], even though one study found a positive non-significant relation [122]. NO_2_ is the most controversial environmental factor studied in the context of PD, with one study finding positive associations [122,123] and another not finding any association [121]. Another study found an association only when adjusting for smoking [122] and one study found it to be both associated and not associated when comparing different cities [120]. Only one study explored sun exposure and found it to be associated when adjusting for age, positively in older subjects, and inversely in subjects below 70 years [124].

Three studies were included in the context of dementia, which comprehended Alzheimer’s, one of which found a positive relation to temperature [125]. The other two studies agreed that NO_2_ and PM_10_ were positively correlated with dementia [126,127]. Furthermore, a study found CO to be inversely correlated with dementia and SO_2_ and O_3_ not to be correlated [126].

Concerning amyotrophic lateral sclerosis (ALS), two studies were found but with no overlapping environmental factors. A study found NO (nitrogen oxide), NO_2_, and Cd (cadmium) to be positively correlated with ALS and O_3_ to be inversely correlated, while benzopyrene was not found to be correlated [128]. The other study found positive correlations between ALS and both precipitation and sun exposure [129].

As for motor neuron disease, in the only study included, it was found to be correlated with Pb [130].

Most studies simply used administrative divisions as geographic analysis to assess exposure to atmospheric factors, but GIS and remote sensing were also broadly used to spatially compute and model those factors. Specifically, studies that used remote sensing focused on NASA’s (National Aeronautics and Space Administration) satellite TOMS (Total Ozone Mapping Spectrometer) to compute the amount of sun exposure [97,101,104,107].

## 4. Discussion

Multiple sclerosis is known to be tied to latitude, thus raising the suggestion that sun exposure, as well as temperature and humidity, are probably the environmental factors behind that relationship [105]. Sun exposure is one of the strongest correlates to latitude [59] and its relation to MS may be explained by biological mechanisms that suggest that ultraviolet radiation may be immunosuppressive [131]. However, these mechanisms seem to be most relevant in early life [60]. The studies retrieved in this systematic review mostly corroborated this hypothesis [98,99,100,101,102,103,104], although three studies did not [105,106,107], with one even attributing the relationship between sun exposure and MS to the high correlation between sun exposure and latitude [105]. Amyotrophic lateral sclerosis also has an environment hypothesis linked to the genetics of the disease and its prevalence. However, studies found in this systematic review focused mostly on water and soil exposures, and are thus outside the scope of this systematic review [61,79]. Another study identified a substantial gap in studies addressing environmental risk factors that may cause ALS [64]. The low incidence rate may be related to the low number of studies concerning the disease [64].

There is evidence that PD is mediated by environmental risk factors such as manganese or lead with the need to better model the exposure geographically model was noted [62]. This corroborates the findings of this systematic review. Dementia risk has been related to occupational exposure to air pollutants such as PM, CO_2_, CO, and SO_2_ [90], corroborating the findings in this systematic review. However, as dementia encloses Alzheimer’s disease, but is not limited to it, no studies were found that approached AD individually, and thus the results might have been slightly different if only AD were studied. The geographic distribution of the included studies shows a lack of studies in South America and Africa, creating the opportunity for research in the field of environmental epidemiology of neurodegenerative diseases in these regions. Analysing the source of data from each study closer, the constatation of the lack of studies from these regions remains. Although Africa has the lowest rates of neurodegenerative diseases in the world, the same cannot be said about South America, so this is a good region to develop further research.

The increasing number of papers published over the years reflects not only the general increase in scientific publications but also the availability of the technology to proceed with the geographic analysis of environmental factors. The rising interest and research investment in this thematic are also reflected in the increased number of studies over the last few years. In particular, papers referring to the new COVID-19 pandemic effects on neurodegenerative diseases have raised awareness of the synergy between this virus and the environment, which increases the risk of developing these diseases. Knowing the risk the atmospheric pollution already represents upon the central nervous system, and by adding the risk of COVID-19, which may penetrate the brain through similar pathways to those of air pollution, there is a higher potential for neuroinflammation and neurodegeneration [39]. Interesting positive relations between NO_2_ levels and COVID-19 fatality rates have been retrieved from a spatial analysis study [132], indicating that this pollutant may weaken the immunity of the lungs, thus allowing the virus to access the bloodstream and eventually reach the brain [133]. Vitamin D has been described as having antiviral properties, and thus the lack of sun exposure may be problematic in people already with neurological conditions, such as MS and PD [134].

One possible limitation of our study might be not having found studies concerning other neurodegenerative diseases, such as Creutzfeldt-Jakob and Huntington’s, therefore not covering the whole domain of neurodegenerative diseases defined at the start of this review. The lack of studies regarding other neurodegenerative diseases is due to the low number of studies using geospatial analysis when considering these diseases. Notwithstanding, most of the included studies focused on MS since it is mostly related to sun exposure—often derived from remote sensing—thus making it the disease most geographically studied. Grey literature was not screened, thus non-indexed papers may have escaped our research. Including this literature could have mitigated the lack of studies focussing on the remaining neurodegenerative diseases that were not found under the scope of this review. As such, further inclusion of this literature could pose an interesting opportunity for future research.

## 5. Conclusions

The most studied neurodegenerative disease in the context of this systematic review was MS, having over half the studies covering it. It is mostly unanimous that sun exposure is a preventive factor in the development of MS. The relationship to air pollutants remains unknown, and further investigation is needed to understand if such a relationship exists. As for the other diseases, air pollutants have become more broadly studied. Metals such as Pb and Mn were found to be related with Parkinson’s disease and ALS, respectively. NO_X_ is the most common pollutant to be studied, but also the most ambiguous, being found to be both related and unrelated with Parkinson’s disease, and positively correlated with dementia and ALS. The geospatial analysis was mostly used to estimate exposure assessment, either by attributing a value to each administrative division and matching it to the residence of the patient or care unit location or by using remote sensing and GIS to compute the atmospheric factor’s concentration or value geographically. The present systematic review provides a basis of the available evidence regarding the influence of environmental factors on neurodegenerative diseases using geospatial analysis. The ever-increasing amount of data and technological possibilities to advance on this analysis supports the development of further research on this topic. It is important to identify mitigation measures to reduce exposure to air pollution. Regulation on the levels of air pollution and accurate monitoring of these same levels could help ensure that the population isn’t over-exposed to harmful components of the atmosphere.

## Figures and Tables

**Figure 1 ijerph-17-08414-f001:**
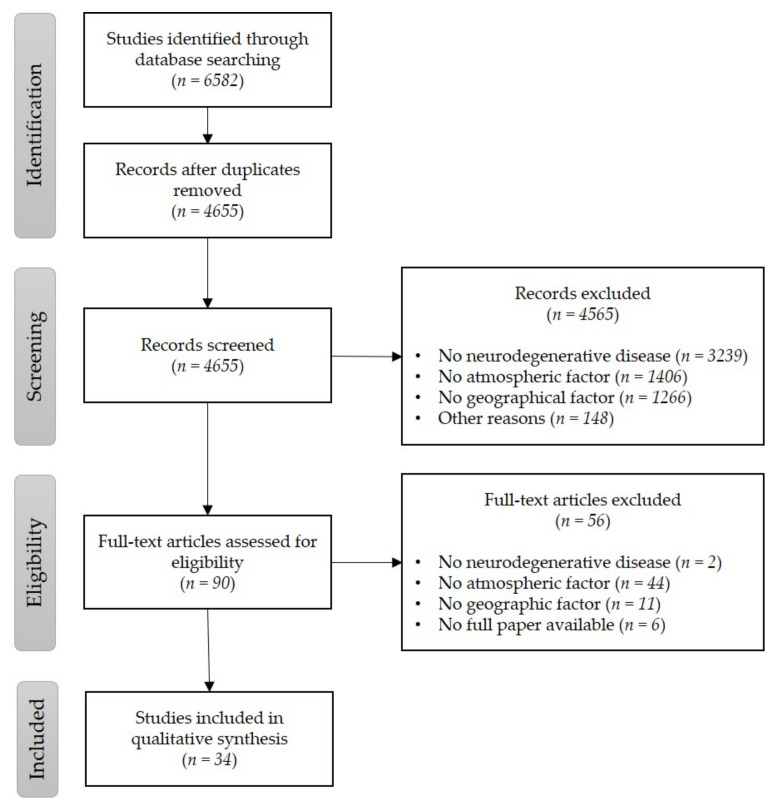
Systematic review PRISMA flowchart.

**Table 1 ijerph-17-08414-t001:** The number of studies addressing each subtopic of the domains, retrieved from the included studies: neurodegenerative diseases, environmental pollutants/factors, and geographic factors.

	Number of Studies	Percentage of Total Entries
Neurodegenerative Disease		
Amyotrophic lateral sclerosis	2	5.9
Dementia (includes Alzheimer’s disease)	3	8.8
Motor neuron disease	1	2.9
Multiple sclerosis	19	55.9
Paediatric multiple sclerosis	2	5.9
Parkinson’s disease	7	20.6
Environmental Pollutant/Factor		
Arsenic (As)	1	1.2
Benzopyrene (C₂OH₁₂)	1	1.2
Benzene (C₆H₆)	1	1.2
Cadmium (Cd)	1	1.2
Carbon monoxide (CO)	5	6.1
Cooper (Cu)	2	2.4
Hydrogen sulphide (H₂S)	1	1.2
Humidity	3	3.7
Index	2	2.4
Magnesium (Mg)	2	2.4
Manganese (Mn)	1	1.2
Nickel (Ni)	1	1.2
Nitrogen dioxide (NOₓ)	10	12.2
Ozone (O₃)	4	4.9
Lead (Pb)	5	6.1
PM₁₀	6	7.3
PM₂₅	8	9.8
Precipitation	6	7.3
Pressure	5	6.1
Radon (Rn)	1	1.2
Sulphur dioxide (SO₂)	6	7.3
Sun exposure	13	15.9
Temperature	8	9.8
Geographic Factors		
Administrative division	16	24.2
Clustering	2	3.0
GIS	14	21.2
Latitude	5	7.6
Longitude	1	1.5
Remote sensing	13	19.7
Residence	14	21.2
Spatial interpolation	1	1.5

**Table 2 ijerph-17-08414-t002:** Study characteristics summary: study design, study limitations, statistical methods, and effect measures.

	Number of Studies	Percentage of Total Entries
Study Design		
Case-control	9	26.5
Cohort	4	11.8
Cross-sectional	12	35.3
Ecological	6	17.6
Methodological	2	5.9
Review	1	2.9
Study Limitations		
Conflict of interests	1	1.1
Confounding	15	16.1
Ecological bias	3	3.2
Exposure assessment	18	19.4
Interpolation	3	3.2
Recall bias	3	3.2
Migration	4	4.3
None is given by the author	5	5.4
Referral bias	2	2.2
Sampling	7	7.5
Statistics	13	14.0
Study design	3	3.2
Survival bias	1	1.1
Time-related	9	9.7
Unassessed patients	6	6.5
Statistical Methods		
ANOVA	2	2.6
Chi-squared	5	6.5
Clustering	2	2.6
Correlation	22	28.6
Cox regression	4	5.2
Linear regression	11	14.3
Logistic regression	10	13.0
None	2	2.6
Poisson regression	3	3.9
Sensitivity analysis	10	13.0
Spatial autoregressive model	1	1.3
T-test	5	6.5
Effect Measures		
Coefficients	14	24.1
Correlation	19	32.8
Hazard ratio	3	5.2
None	3	5.2
Odds ratio	11	19.0
Prevalence	4	6.9
Relative risk	4	6.9

**Table 3 ijerph-17-08414-t003:** Study characteristics of the 34 included papers [97].

Ref	Country (year)	Neurodege-Generative Disease	Environmental Factor	Geographic Factor	Study Design	Study Limitations	Statistical Methods	Outcome
[97]	Canada (2012)	Multiple sclerosis	Sun exposure	GIS, Remote Sensing	Methodological	Exposure assessment	None	None
[98]	Israel (1971)	Multiple sclerosis	Sun exposure, Temperature, Precipitation, Humidity	Residence	Review	None given by the authors	None	None
[99]	Bulgaria (1987)	Multiple sclerosis	Sun exposure, Temperature, Precipitation	Administrative division, Latitude	Cross-sectional	Unassessed patients	Correlation, Chi-squared, Linear regression	Correlation, Coefficients
[100]	Australia (2001)	Multiple sclerosis	Sun exposure, Temperature, Precipitation	Administrative division, Latitude, Remote Sensing	Ecological	Confounding, Exposure assessment	Correlation, Poisson regression	Prevalence, Correlation
[101]	Canada (2011)	Multiple sclerosis	Sun exposure	Latitude, Longitude, Remote Sensing	Cross-sectional	None given by the authors	Correlation, Linear regression	Correlation
[102]	England (2011)	Multiple sclerosis	Sun exposure	GIS, Remote Sensing	Cross-sectional	Confounding, Sampling, Statistics	Correlation, Linear regression	Correlation, Coefficients
[103]	USA (2017)	Multiple sclerosis	Sun exposure, Temperature	Administrative division, GIS, Remote Sensing	Cross-sectional	Confounding, Statistics	Correlation, Linear regression	Correlation, Coefficients
[104]	USA (2018)	Multiple sclerosis	Sun exposure	Residence, Remote Sensing	Cohort	Confounding, Exposure assessment, Interpolation, Recall bias, Migration, Survival bias, Time related	Cox regression	Relative risk, Hazard ratio
[105]	USA (1983)	Multiple sclerosis	Sun exposure, Temperature, Precipitation, Humidity	Latitude	Case-control	None given by the authors	Logistic regression	Relative risk
[106]	Italy (2016)	Multiple sclerosis	Sun exposure	Administrative division, GIS	Cross-sectional	Confounding, Ecological bias, Time related	Correlation, Linear regression	Correlation, Odds ratio
[107]	Canada (2018)	Multiple sclerosis	Sun exposure	Residence, Remote Sensing	Cohort	Confounding, Exposure assessment, Time related	Linear regression	Coefficients
[108]	Norway (2010)	Multiple sclerosis	Sun exposure, Temperature, Precipitation	Administrative division	Cross-sectional	Migration, Statistics	ANOVA, Poisson regression	Prevalence
[109]	Italy (2018)	Multiple sclerosis	PM_2.5_	Residence, Remote Sensing	Cross-sectional	Conflict of interests, Confounding, Study design	Correlation, Chi-squared	Correlation, Coefficients
[110]	USA (2008)	Multiple sclerosis	PM_10_, PM_2.5_, NO_X_, SO_2_, CO	Administrative division	Cross-sectional	None given by the authors	Correlation, T-test, Linear regression	Correlation, Coefficients
[111]	Italy (2005)	Multiple sclerosis	SO_2_	Administrative division, Latitude	Cross-sectional	Exposure assessment, Interpolation	Correlation, Linear regression	Correlation, Coefficients
[112]	Iran (2014)	Multiple sclerosis	PM_10_, NO_X_, SO_2_	Clustering, GIS	Cross-sectional	Confounding, Statistics, Study design	Correlation, Clustering	Correlation, Coefficients
[113]	Iran (2018)	Multiple sclerosis	Index	Administrative division, GIS, Residence	Cross-sectional	Exposure assessment, Statistics	Correlation, Logistic regression	Odds ratio, Coefficients
[114]	Norway (1997)	Multiple sclerosis	Mg	Administrative division	Methodological	Confounding	Correlation	None
[115]	England (2016)	Multiple sclerosis	Rn	Residence	Ecological	Sampling, Statistics, Unassessed patients	Correlation, Chi-squared, Linear regression	Correlation, Coefficients
[116]	USA (2017)	Paediatric Multiple sclerosis	Index	GIS, Residence	Case-control	Exposure assessment, Statistics, Time related, Unassessed patients	Logistic regression	Odds ratio, Coefficients
[117]	USA (2018)	Paediatric Multiple sclerosis	PM_10_, PM_2.5_, NO_X_, SO_2_, CO, O_3_, Pb	Administrative division, GIS, Residence	Case-control	Exposure assessment, Referral bias, Time related	T-test, Logistic regression	Odds ratio
[118]	USA (2010)	Parkinson’s disease	Cu, Pb, Mg	Administrative division	Ecological	Confounding, Exposure assessment, Statistics	Logistic regression, Sensitivity analysis	Relative risk, Odds ratio
[119]	Spain (2016)	Parkinson’s disease	Pb	Administrative division, GIS	Ecological	Exposure assessment, Sampling, Unassessed patients	Correlation, T-test	Correlation, Coefficients
[120]	Canada (2007)	Parkinson’s disease	NO_X_, Mn	Residence, Remote Sensing, Spatial interpolation	Case-control	Confounding, Exposure assessment, Interpolation, Study design, Time related	Correlation, Linear regression, Logistic regression, Cox regression, Sensitivity analysis	Prevalence, Correlation, Odds ratio
[121]	USA (2016)	Parkinson’s disease	PM_10_, PM_2.5_, NO_X_	GIS, Residence	Case-control	Exposure assessment, Recall bias, Statistics, Time related	Correlation, Logistic regression, Sensitivity analysis	Correlation, Odds ratio
[122]	Australia (2020)	Parkinson’s disease	PM_2.5_, NO_X_	Residence, Remote Sensing	Cross-sectional	Recall bias, Referral bias, Sampling	Logistic regression, Sensitivity analysis	Odds ratio
[123]	Taiwan (2016)	Parkinson’s disease	NO_X_, CO	GIS, Residence	Case-control	Confounding, Sampling, Statistics	Correlation, Chi-squared, Logistic regression, Sensitivity analysis	Correlation, Odds ratio
[124]	France (2017)	Parkinson’s disease	Sun exposure, PM_2.5_	Administrative division, Remote Sensing	Ecological	Ecological bias, Exposure assessment, Migration	Correlation, Poisson regression, Sensitivity analysis	Correlation, Relative risk
[125]	USA (2019)	Dementia	Temperature	Administrative division, Residence, Remote Sensing	Cohort	Confounding, Exposure assessment, Statistics	Correlation, Cox regression, Sensitivity analysis	Correlation, Hazard ratio
[126]	Taiwan (2019)	Dementia	PM_10_, NO_X_, SO_2_, CO, O_3_	Clustering	Case-control	Confounding, Exposure assessment, Statistics, Unassessed patients	Correlation, Logistic regression, Sensitivity analysis	Odds ratio
[127]	Canada (2017)	Dementia	PM_2.5_, NO_X_, O_3_	GIS, Residence, Remote Sensing	Cohort	Confounding, Exposure assessment, Time related, Unassessed patients	Cox regression, Sensitivity analysis	Hazard ratio
[128]	Spain (2018)	Amyotrophic lateral sclerosis	PM_10_, PM_25_, NO_X_, SO_2_, CO, O_3_, Cu, Pb, As, Ni, Cd, C_6_H_6_, H_2_S, C_6_OH_12_	GIS	Case-control	Ecological bias, Exposure assessment, Sampling, Statistics	T-test, Chi-squared, Linear regression, Sensitivity analysis	Prevalence, Odds ratio
[129]	Taiwan (2013)	Amyotrophic lateral sclerosis	Sun exposure, Temperature, Precipitation, Humidity, Pressure	Administrative division	Case-control	Exposure assessment, Migration, Sampling, Time related	Correlation, Spatial autoregressive model, Clustering	Correlation, Coefficients
[130]	Spain (2016)	Motor neuron disease	Pb	Administrative division, GIS	Ecological	None given by the authors	Correlation, T-test, ANOVA	Correlation, Coefficients

**Table 4 ijerph-17-08414-t004:** Studies published by each geographic region, grouped by neurodegenerative disease.

	Amyotrophic Lateral Sclerosis	Dementia	Motor Neuron Disease	Multiple Sclerosis	Paediatric Multiple Sclerosis	Parkinson
Asia	[129]	[126]				[123]
Australia				[100]		[122]
Europe	[128]		[130]	[99,102,106,108,109,111,114,115]		[119,124]
Middle East				[98,112,113]		
North America		[125,127]		[97,101,103,104,105,107,110]	[116,117]	[118,120,121]

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
