# Peer review of "Geospatial Analysis of Environmental Atmospheric Risk Factors in Neurodegenerative Diseases: A Systematic Review"

_ijerph, 2020, doi:10.3390/ijerph17228414_

Round 1

Reviewer 1 Report

1、 For the topic,there are no specific Geospatial analysis in the results and discussion.
2、 For a meta analysis, the major consideration is the data capacity. In the study, there are 34 papers in total, is it enough to reflect how does environmental risk factors influence the neurodegenerative diseases?
3、 The causality between environmental risk factors and neurodegenerative diseases should be explained by mechanism. A brief introduction will be better.
4、 Alzheimer’s disease (AD) is the most prevalent neurodegenerative disease worldwide, why there are no studies shown in the results and discussion.
5、 Why exclude soil and water pollutants? Generally, these two pollutants have far more influences for people’s health.
6、 There is no need to specify the screening process. For the Figure 1 flowchart, it is no necessary, you can paraphrase it into several sentences.

Reviewer 2 Report

I have carefully reviewed the manuscript with ID: ijerph-968799 , the authors present A systematic review of geospatial analysis of environmental risk factors in neurodegenerative diseases, which may be of interest to the IJERPH community, however, below I list some suggestions to improve the quality of the manuscript , and clarify its main contributions:

Q1. At the end of the Introduction, it is suggested to write how the manuscript is organized, this with the purpose of giving an overview of the content.

Q2. Section 2, Materials and Methods, It is suggested to specify the date range used for the database search, i.e. start and end date, as part of the methodology employed.

Q3. Line 130, Error! Reference source not found.

Q4. Line 139-140, Error! Reference source not found.

Q5. Table 1-3. In section 3, it is necessary to add a descriptive paragraph of the results shown in Table 1-3. (All tables and figures must be referenced within the document). That is, the authors should write a brief discussion of the main results shown in the tables.

Q6. It is necessary to add a descriptive paragraph of the results shown in Figure 1

Q7. Line 146-151, line 164, Error! Reference source not found.

Q8. Line 215, [42].Amyotrophic….. Put space after the dot.

Q9. According to the comments on line 38-39, Alzheimer’s disease (AD) is the most prevalent neurodegenerative disease worldwide [5], however, in Table it does not appear, why it was not included?

Q10. State of the art, I consider that it is necessary to carry out a deep search of the articles recently reported in the literature, particularly in 2020, because only two Articles from 2020 were found in the References, and 2 from Organization, W.H. For example, currently, some works already report neurodegenerative diseases due to Sars-CoV-2 or Covid 19 pandemic, which is a great public health problem worldwide, here I share some examples, however in the literature more related work can be found.
DOI: 10.1016/j.jns.2020.117081
DOI: 10.1007/s10072-020-04575-3
DOI: 10.3390/brainsci10090612
PubMed ID: 32798347, link: https://www.scopus.com/inward/record.uri?eid=2-s2.0-85089542665&partnerID=40&md5=d86994e713a96249317ae114fb5f3cc4

Q11. In the Conclusions, it is necessary to emphasize the main results related to the geospatial analysis and the main environmental risk factors in degenerative diseases. Also, it would be interesting to mention the main preventive actions to decrease the risks of neurodegenerative diseases that have been found in the literature.

Reviewer 3 Report

Excellent quality of presentation, in both contents and language style. The idea of this systematic review is to be lauded andhas been thoroughly researched and presented, even though, the title and abstract are a little bit misleading. It should be made clear in the abstract that ONLY atmospheric factors are considered. Just so the limitations of this systematic review are clear from the start, and is consistent with the discussion. In fact it is not clear why they only focused on atmospheric factors, after initially including everthing in the search, and then excluding so many papers. This becomes even more confusing, when they initially note as «inconclusive» the link between ALS and atmospheric environmental factors, but then concede that clear links have been found between ASL and water and soil. This lack of clarity on the «limited scope of the study» also weakens the argument that «there is a low number of studies on degenerative diseases using geospatial analysis».  It is in fact a bit odd to include so many terms and «or» options in the initial search, which produces 6482 (not clear in the text what is meant by duplicates) and then end up with only 34 studies. Finally, and this is probably just an oversight, the numbering starts at 9 but also ends at 40. Are the first 8 missing or is there a miscounting issue (34 or 36)?

Furthermore, they initially say that they didn’t pay attention to language, but it’s clear that the searches will pull out studies published in English. This is understandable, but to then note that «more studies should be done in those areas» (South America and Africa), is rather naive and surprising. I can imagine that if a similar literature search is performed in Spanish, Portuguese, and French just to name a few, the above mentioned geographical areas would be better represented.

All this does not take away from the value and publishability of this paper. As stated initially, I am very impressed by the scope, contents and presentation of this paper.

Reviewer 4 Report

This is a reasonable start to a systematic review of the relationship between atmospheric pollution and sunlight, and neurodegenerative diseases. There are several things that need to be addressed and included before it can be accepted.

  • The title is not accurate: all environmental risk factors were not considered. Only atmospheric pollutants and sunlight. This needs to be stated correctly everywhere in the manuscript, including Figure 1. Or the authors could expand the scope to consider all environmental risk factors.
  • The writing needs to be checked for English language editing.
  • The abstract should include quantitative data to back up the statements that are made. A range of effect measures is acceptable across the studies.
  • Lines 65-76. Authors need to be very explicit here. Is geospatial analysis anything that uses geospatial tools in exposure assessment or the epidemiologic analysis? Or only one or the other? Be specific and explicit.
  • Lines 90-92. Authors titles this article “geospatial” but they didn’t look up “geospatial” as one of their search terms? That seems like a very odd decision. I would expand the search terms to also include space-time, spatio-temporal, and geospatial.
  • I am not familiar with the term memory bias. Did you mean recall bias?
  • Line 137: Remove “However”.
  • There should be a geospatial component to the qualitative synthesis. Some discussion of how geospatial considerations were made across the studies.
  • There should be a quantitative summary. Typically showing ORs or RRs from the studies in a forest plot.

Round 2

Reviewer 1 Report

  • Don’t start sentence with Arabic numerals in thesis.
  • In the Abstract, We searched PubMed, Web of Science and Scopus for all 20 scientific studies considering the following three domains: neurodegenerative disease, 21 environmental atmospheric factor, and geographical analysis. This sentence is a repetition of the Keywords, it is suggested to delete it and present it in the Materials and Methods. For the conclusion, it will be much more clear to demonstrate like this: the meta-analysis indicates:(1)…(2)…(3)…
  • For the last paragraph of the Introduction, the presentation like:In section 2,3,4,5,these are kind of boring and wordiness. It is suggested to put emphasis on the research objectives.
  • For the Results, the identification and screening process should be put it into Materials and Methods. For the lats paragraph of Results, Specifically, studies that used remote sensing focused on NASA’s satellite TOMS to compute the amount of sun exposure. Which study? There is no reference.
  • For the Conclusions, it is abrupt to state the new COVID-19 pandemic effects on the neurodegenerative diseases, it is suggested to add some information about the new COVID-19 in the Introduction.

Reviewer 2 Report

I have thoroughly reviewed the revised version of the manuscript and I see that the authors have correctly addressed the comments and suggestions. Thank you

Author Response

Dear Reviewer,

The authors would like to thank you again for reviewing the manuscript “Geospatial analysis of environmental risk factors in neurodegenerative diseases: A systematic review”. We are pleased we answered all your questions. Please contact us if you have any further questions or comments. Thank you.

Reviewer 4 Report

Authors adequately addressed my concerns. Quality of English is still a concern. Overall impact of this work is likely to be limited given the lack of quantitative findings. 

Author Response

This manuscript is a resubmission of an earlier submission. The following is a list of the peer review reports and author responses from that submission.